# Antimicrobial and Toxicity Evaluation of Imidazolium-Based Dicationic Ionic Liquids with Dicarboxylate Anions

**DOI:** 10.3390/pharmaceutics13050639

**Published:** 2021-04-29

**Authors:** Bruna L. Kuhn, Taís F. A. Kaminski, Ânderson R. Carvalho, Alexandre M. Fuentefria, Bianca M. B. C. Johann, Edilma E. da Silva, Gustavo P. Silveira, Tássia L. da Silveira, Félix A. A. Soares, Nilo Zanatta, Clarissa P. Frizzo

**Affiliations:** 1Department of Chemistry, Federal University of Santa Maria, Santa Maria 97105-900, Brazil; brunakuhn.quimica@gmail.com (B.L.K.); nilo.zanatta@ufsm.br (N.Z.); 2Laboratory of Applied Mycology, School of Pharmacy, Federal University of Rio Grande do Sul, Porto Alegre 90470-440, Brazil; tais_farmacia@hotmail.com (T.F.A.K.); andersonrc87@gmail.com (Â.R.C.); alexandre.fuentefria@ufrgs.br (A.M.F.); 3Postgraduate Program in Agricultural Microbiology and the Environment, Instituto de Ciências Básicas da Saúde-Campus Centro, Rua Sarmento Leite 500, Porto Alegre 90050-170, Brazil; biancabjohann@gmail.com (B.M.B.C.J.); 00211191@ufrgs.br (G.P.S.); 4Postgraduate Program in Chemistry, Instituto de Química-Campus Vale, Av. Bento Gonçalves 9500, Porto Alegre 91501-970, Brazil; edilmaelayne@gmail.com; 5Departament of Biochemistry and Molecular Biology, Federal University of Santa Maria, Avenida Roraima 1000, Santa Maria 97105-900, Brazil; tassiasilli@yahoo.com.br (T.L.d.S.); felix@ufsm.br (F.A.A.S.)

**Keywords:** dicationic ionic liquid, dicarboxylate anion, antimicrobial activity, *C. elegans*, cytotoxicity, ESKAPE panel

## Abstract

Imidazolium-based dicationic ILs (DILs) presenting antimicrobial activity and relatively low toxicity are highly desirable and are envisioned for use in live tissue to prevent bacterial or fungal infections. In this context, we present here DILs with dicarboxylate anions [C_n_(MIM)_2_[C_n_(MIM)_2_][CO_2_-(CH_2_)_m_CO_2_], in which *n* = 4, 6, 8, and 10, and m = 0, 1, 2, 3, 4, and 5. The results showed that DILs with an alkyl chain spacer of ten carbons were active against yeasts and the bacterial strains tested. However, most of the DILs were cytotoxic and toxic at 1 mM. By contrast, DILs with alkyl chains possessing less than ten carbons were active against some specific *Candidas* and bacteria (mainly *S. aureus*), and they showed moderate cytotoxicity. The best activity against Gram-positive bacteria was observed for [C_4_(MIM)_2_][Pim] toward MRSA. For the DILs described herein, their level of toxicity against *C. elegans* was lower than that of most of the mono- and dicationic IL analogs with other anions. Our results showed that the presence of carboxylate anions reduces the toxicity of DILs compared to DILs containing halide anions, which is particularly significant to the means of designing biologically active compounds in antimicrobial formulations.

## 1. Introduction

Given the possible applications in drug synthesis, drug delivery systems, and biomedical materials, the antimicrobial, cytotoxic, and toxic properties of ionic liquids (ILs) have attracted significant attention from medical scientists. A diversity of measures of antimicrobial activity has been reported for all classes of ILs to evaluate the antimicrobial response of a widespread range of microorganisms [1,2]. The mechanisms of antimicrobial activity in ILs have not yet been fully elucidated; however, it has been largely described that the alkyl chains are the protagonist, most likely by disturbing the chemical structure of the biological membranes [2,3,4]. The highest antimicrobial activity was observed in ILs with an alkyl chain length of 12 or 14 carbons. Still, a marked reduction in antimicrobial activity for IL with alkyl chains with more than 16 or less than 10 carbons was reported [2,3,4]. Regarding the chemical structure, the presence of polar groups in the hydrophilic cations in imidazolium-based ILs could also suggestively disrupt their antimicrobial activity [3,4,5,6]. Imidazolium-based dicationic ILs (DILs) are a kind of IL that has been receiving the attention of the scientific community over the last decade, ever since its lower toxicity compared to monocationic ILs was described [7]. The toxicity of ILs has been shown to be moderate by the presence of polar groups such as amide, ester, carboxyl, and hydroxyl in the cationic portion. This fact also improved the biodegradability of ILs [8,9]. DILs presenting antimicrobial activity and relatively low toxicity toward mammals are highly desirable and are envisioned for use in live tissue to prevent bacterial or fungal infections [10]. The majority of DILs described in the literature have a halide anion and a long alkyl substituent to balance lipophilicity for the appropriate bioavailability of drug candidates. Additionally, bioactivity can be improved via the introduction of an organic natural anion [11]. The toxic and irritant effects of molecules containing halide anions can be reduced by these structural changes, which is mostly important in the proposal of biologically active compounds. Additionally, the biocompatibility and biodegradability of ILs can also be enhanced because of the insertion of natural anions [11].

A group of six bacteria is recognized as having escaped the biocidal action of drugs and is responsible for most of the nosocomial infections worldwide. Thus, the acronym ESKAPE is used to refer to this panel, which includes the following strains: *Enterococcus faecium, Staphylococcus aureus, Klebsiella pneumoniae, Acinetobacter baumannii, Pseudomonas aeruginosa*, and *Enterobacter* spp. [1]. Most of these bacteria are multiresistant, which makes it imperative to discover new drugs to combat infections caused by these pathogens [12]. *Candida albicans* has been the most common fungal strain isolated from hospitalized patients; however, an increasing number of other infections have been reported from another *Candida* spp., such as *C. krusei, C. tropicalis, C. glabrata, C. guilliermondii, and C. parapsilosis.* Azoles are generally prescribed to treat infections caused by these yeasts [13]; however, due to resistance, other classes of antifungals and their action mechanisms need to be discovered [12].

There are a few reports about cellular toxicity of imidazolium-based DILs [10,14]. Moreover, the biodegradability of these ILs has grown due to their potential ecological impacts [15,16]. Indeed, the environmental impact of imidazolium-based DILs is not well-known, which could be a limitation to the concrete use of ILs in biological and biomedical applications.

*Caenorhabditis elegans* (*C. elegans*) is a mud nematode that also lives in aquatic habitats. This habit makes it appropriate for chemical exposure in aqueous media [17]. Some advantages of *C. elegans* as a model organism for toxicological investigations are that it is a satisfactory categorized genome, its link to human biology, its ease of handling and conservation, and its short and prolific life cycle [18]. Thus, the survival of these nematode species is a suitable environmental and ecological marker of activity of ILs. Previous, imidazolium-based DILs [C_8_(MIM)_2_][2X], in which X = Cl, Br, NO_3_, SCN, and BF_4_, and their toxicological effects against *C. elegans* were reported by us. In our early scanning, increasing concentrations of IL (0.01, 0.1, 1.0, and 10 mM) were added to the first stage of the worms (larval stage), which is more sensitive, and remained for 1 h. Results showed that the survival of above 90% of *C. elegans* indicated that these ILs do not have toxic effects at the concentrations evaluated [19].

We recently described the synthesis and thermophysical properties of imidazolium-based DILs with dicarboxylate counterions as anions—[C_n_(MIM)_2_[C_n_(MIM)_2_][CO_2_-(CH_2_)_m_CO_2_], in which *n* = 4, 6, 8, and 10, and m = 0 (oxalate), 1 (malonate), 2 (succinate), 3 (glutarate), 4 (adipate), and 5 (pimelate) (Figure 1) [20].

Considering that these DILs can be used as multifunctional materials in dental implants [21], herein we present the screening of 18 examples of [C_n_(MIM)_2_[C_n_(MIM)_2_][CO_2_-(CH_2_)_m_CO_2_] against the ESKAPE panel of bacteria and *Candida* spp., together with their cytotoxicity and *C. elegans* toxicity.

## 2. Materials and Methods

### 2.1. Synthesis of DILs

DILs were synthesized in accordance with the method of a previous work [20], and it is described here for [C_4_(MIM)_2_][C0]: 1,4-Bis(3-methylimidazolium-1-yl)butane hydroxide was prepared from 1,4-Bis(3-methylimidazolium-1-yl)butane bromide ethanolic solution, using anion exchange resin (Amberlite IRN 78), and the reaction was monitored using AgNO_3_ 0.1 M solution. An ethanolic solution of 1,4-Bis(3-methylimidazolium-1-yl)butane hydroxide was subsequently added dropwise to an equimolar ethanolic solution of oxalic acid, and the mixture was stirred at 25 °C for 24 h. The solvent was then evaporated under reduced pressure, washed with diethyl ether (10 mL, twice), and dried under a vacuum for 72 h at 60 °C. The structures of the resulting DILs were confirmed and characterized by ^1^H and ^13^C NMR spectroscopy, mass spectrometry, thermal analysis (TGA and DSC), and IR spectroscopy [20].

### 2.2. Antibacterial Activity

*Escherichia coli* (ATCC 25922), *Enterococcus faecalis* (ATCC 29212 and ATCC 51299), and the ESKAPE panel of pathogens, including strains of methicillin-resistant (ATCC 33591) and methicillin-susceptible (ATCC 25923) *S. aureus,* as well as *K. pneumoniae* (ATCC700603), *P. aeruginosa* (ATCC 27853), *A. baumanni* (ATCC 19606), *A. baumanni* (IOC 3174), and *E. aerogenes* (ATCC 13048), were obtained by donation from Instituto Oswaldo Cruz (RJ, Brazil). Following the application of the CLSI microdilution method (CLSI 2019), using BBLTM [22] and Mueller Hinton II broth (Interlab, Brazil), the minimum inhibitory concentrations (MICs) were evaluated. Two-fold serial dilutions of each DIL (5 mM stock solution) were prepared in 96-well ELISA plates, and then inoculated with 5 × 10^5^ CFU·mL^−1^ of the bacterial suspension in triplicate. Plates were incubated at 37 °C for 16–20 h. The ampicillin and imipenem (purity of 97%) antibiotics, which were used as controls, were purchased from Sigma-Aldrich (Sao Paulo, Brazil).

### 2.3. Antifungal Activity

#### 2.3.1. Susceptibility Test

The MIC of the DILs and fluconazole (used as control) against the *Candida* species were determined by the broth microdilution method, in accordance with the M27-A3 protocol [23]. Serial dilutions were made in RPMI 1640 medium, and the experiments were done in triplicate. The MIC was defined as the lowest concentration of compound in which the microorganisms tested—*C. albicans* (CA 02), *C. krusei* (CK03), *C. parapsilosis* (CPRL 38), and *C. tropicalis* (CT 08)—did not show visible growth at 24 and 48 h.

#### 2.3.2. Cell Culture

Cell culture was prepared using venous blood collected by venipuncture from a young adult volunteer with more than 18 years of no medication use. The leukocytes (protocol approved by the Ethics Committee of Universidade Federal do Pampa, under number 27045614.0.0000.5323), which were obtained by centrifugation gradient, were immediately transferred to the culture medium containing RPMI 1640 medium, supplemented with 10% fetal bovine serum and 1% streptomycin/penicillin. The cell culture flasks were placed in an oven at 37 °C for 72 h. The negative control was prepared with PBS 7.4 buffer, while the positive control was prepared with hydrogen peroxide [24].

#### 2.3.3. Cell Viability

In accordance with the work of Burow et al. [25], cell viability was assessed by loss of leukocyte membrane integrity, using trypan blue dye. The technique uses leukocytes that are subjected to the trypan blue reagent and, after three minutes, an aliquot is placed in a Neubauer chamber for viewing under a microscope (400× magnification). Hydrogen peroxide was used as a positive control. Unviable cells acquire a blue color; thus, they are visually differentiated from viable cells. A total of 100 cells were counted.

### 2.4. C. elegans Strains: Maintenance and Treatment

Wild-type N2 worms were obtained from the Caenorhabditis Genetics Center (CGC, University of Minnesota, Minneapolis, MN, USA) and kept at 20 °C. Wild-type hermaphrodite worms were cultured on NGM plates (1.7% agar, 2.5 mg.mL^−1^ peptone, 25 mM NaCl, 50 mM KH_2_PO_4_ pH 6.0, 5 μg mL^−1^ cholesterol, 1 mM CaCl_2_, 1 mM MgSO_4_) with fresh *E. coli* OP50 until the worms reached the adult with eggs stage [26]. The N2 eggs were obtained by isolating embryos from gravid hermaphrodites using bleaching solution (1% NaOCl, 0.25 M NaOH), which were then washed three times and stored overnight in M9 buffer (42 mM Na_2_HPO_4_, 22 mM KH_2_PO_4_, 8.6 mM NaCl, and 1 mM MgSO_4_) in order to obtain all animals in stage L1. The L1 population was used to perform the survival assay. The L1 stage worms were exposed to IL (0.001, 0.01, 0.1, 1, and 10 mM) for 1 h, and then transferred to 1.5 mL conical tubes containing M9 buffer (42 mM Na_2_HPO_4_, 22 mM KH_2_PO_4_, 8.6 mM NaCl, and 1 mM MgSO_4_), together with each respective DIL. The worms were continuously shaken to stimulate oxygenation. After 1 h, the worms were washed three times with M9 buffer, and the survival assay was then performed [19].

#### Survival Assay

After exposure to different concentrations of each DIL, the worms were transferred to microscopic slides in the M9 buffer. About 100 worms were analyzed per group per experiment, and the number of survivors was quantified. Animals were considered dead when they did not respond to a tactile stimulus. The survival assay was repeated four times in triplicate [27].

## 3. Results

### 3.1. Antifungal Activity and Cell Viability

In this study, fluconazole breakpoints were used for the *Candida* species according to CLSI. Comparatively, it is understood that the strains used in our susceptibility test have a resistance profile when treated with fluconazole. *C. albicans* and *C. tropicalis* parallel MIC of 4 μg·mL^−1^ and as an SDD interpretation (sensitive dose-dependent), whereas the *C. parapsilosis* strain showed itself with the MIC of 8 μg·mL^−1^ interpreted as resistant to fluconazole. Finally, *C. krusei* has no established breakpoint; however, it is assumed that there is intrinsic resistance of this species to fluconazole.

The results showed that, after 24 h, all DILs were active in concentrations below 1 mM. Except for [C_8_(MIM)_2_][Glu], MIC values were greater than 0.312 mM for *C. albicans* and *C. parapsilosis*, which indicates a weak antifungal activity of the DILs tested against these strains (see Table 1). The best MIC values (0.039–0.078) were obtained for *C. krusei* and *C. tropicalis*; however, [C_8_(MIM)_2_][Mal] and [C_8_(MIM)_2_][Oxa] were not active against *C. krusei* and *C. tropicalis*, respectively. However, we noticed a trend where only [C_8_(MIM)_2_][Glu] was able to inhibit *C. albicans* and *C. parapsilosis*, thus becoming a better choice among DILs. Therefore, we can say that there is an influence of the carbon length in the anion of the [C_8_(MIM)_2_]^2+^ DILs with the MIC.

The cationic effect on antifungal activity was evaluated by varying the dicationic spacer chain, keeping the succinate anion as a counter ion ([C_n_(MIM)_2_][Suc], in which *n* = 4, 6, 8, 10)—see Table 2. DILs with spacer chains of six and 10 methylenes had the best overall antifungal activity with the lowest MICs. Where [C_10_(MIM)_2_][Suc] had lower MICs for most of the *Candida* species tested, then we had [C_6_(MIM)_2_][Suc] with an equally good performance, but lower when compared to its analog with a 10-methylene spacer. The results show that there was an effect when there was an increase in the spacer chain of the dicationic ion, modifying the necessary concentration of IL to prevent the growth of fungi.

Based on the MICs of DILs against *Candidas* (Table 2), and using the trypan blue dye, two concentrations were chosen to determine cell viability by decreasing the integrity of the leukocyte membrane—see Figure 2 [25]. At the concentrations tested, there was no significant decrease in cell viability of the DILs compared to the negative control (>80% for the highest concentration tested). The lowest cell viability (about 70%) was for [C_10_(MIM)_2_][Glu], followed by [C_8_(MIM)_2_][Pim] and [C_10_(MIM)_2_][Suc]. These results show that in MICs for fungi, DILs do not cause damage to the membrane.

The toxicity (log EC50, μM) for DILs is lower when compared to monocationic ILs, as reported by Montalbán, Víllora, and Licence [7]. Comparing the same ILs that have alkyl chains with the same number of carbons, [C_8_(MIM)]^+^ (monocationic—side chain) and [C_8_(MIM)_2_]^2+^ (dicationic—spacer chain), these have EC50 values 2.34 and 0.71 μM, respectively, showing three times less toxicity for the dicationic analog. In the same work, it was evidenced that the increase in the alkyl chain promotes an increase in toxicity—a fact also evidenced in the work of Stolte et al. [28]. Regarding the imidazole head, the insertion of an additional polar head, when comparing mono and dicationic ILs, leads to a reduction in toxicity [7,10]. It has also been shown that the toxicity of LIs depends mainly on the cationic chain, with a lesser impact on anions [28].

Small molecules that have small alkyl chains—resulting in low molecular volumes—make deep insertions in the lipid bilayer, causing minor disturbances as explained by Gal et al. [29]. Lim et al. [30] suggest that small alkyl chains, when inserted into the lipid membrane, create curvatures in their disposition, which leads to a narrowing of the lipid bilayer and, as a consequence, its destabilization [30]. In a previous work, we demonstrated that in solution, DILs based on imidazole with bromide anion, with different alkyl spacer chains, have different structural organizations at the water–air interface [31]. Thus, we can make a relationship with the behavior of DILs derived from carboxylic acids and their behavior at the plasma–membrane interface. Alkyl chains with a greater amount of methylene in their structure show a strong fixation of the molecule on the surface of the lipid bilayer. When the alkyl chain is long, the molecule folds like a hairpin and can again anchor itself to the membrane (Figure 3). This fact most likely occurs giving rise to a considerable disturbance in the membrane, consistent with the effect of micellization.

Considering the hydrophobicity of ILs, when comparing monocationic and dicationic molecules, the latter have less hydrophobicity—dicationic compounds have negative log P from −4.54 to −7.24, while their monocationic analogs have log P from −0.37 to 2.97 [7]. This directly affects the permeability of the molecules, considerably reducing their toxicity, since log P values between 1 and 5 are able to accumulate in the membranes and cause changes in fluidity by changing their structure and function [32]. Therefore, the greater the hydrophobicity, the greater the accumulation in the plasma membrane, causing loss of integrity and ionic dysregulation. In our study, we also performed hydrophobicity prediction calculations (log P) using the Chemdraw 12 and Molinspiration software (Table 3).

Therefore, there is no conflict between cytotoxicity and antifungal activity for most DILs, which indicates that they are candidates for further studies that could lead to biological applications. This means that, in contact with both fungal cells and host cells, ILs can limit fungal growth, but not the proliferation of host cells.

### 3.2. Antibacterial Activity

The MICs of the DILs were determined against the ESKAPE panel of bacteria: Gram-positive strains of methicillin-sensitive *S. aureus* (MSSA-ATCC 25923), of methicillin-resistant *S. aureus* (MRSA-ATCC 33591), of *E. faecalis* (ATCC 29212, ATCC 51299), and of *E. faecium* (ATCC 6569)—see Table 4; and Gram-negative strains of *A. baumannii* (ATCC 19606, IOC 3174), of *K. pneumoniae* (ATCC 700603), of *P. aeruginosa* (ATCC 27853), and of *E. aerogenes* (ATCC 13048). Although *E. coli* (ATCC 25922) is not present in the ESKAPE panel, this strain has been chosen in the past as a model for fast identification of molecules against Gram-negative bacteria—see Table 5. Since at least one DIL from each length group was active against *E. coli* at the breakpoint chosen (2.5 mM), it was decided to screen all DILs against the whole Gram-negative ESKAPE panel (Table 4).

The best activity against Gram-positive bacteria was observed for [C_4_(MIM)_2_][Pim]. This DIL had an MIC of 78 µM toward MRSA. However, three-fold serial dilution was obtained for this DIL against MSSA (625 µM), and there was no activity toward the other Gram-positive and Gram-negative pathogens screened. On the other hand, [C_4_(MIM)_2_][Adi] and [C_10_(MIM)_2_][Pim] were active against the whole Gram-positive panel tested, with MICs of 0.625–1.25 mM. [C_4_(MIM)_2_][Adi] was also active against the Gram-negative strains *E. Coli* (312 µM), *P. Aeruginosa* (2.5 mM), and *E. faecium* (625 µM). Similarly, [C_10_(MIM)_2_][Pim] was also active toward *E. Coli* (625 µM), *K. pneumoniae* (625 µM), *P. Aeruginosa* (312 µM), *E. faecium* (156 µM), and *E. aerogenes* (156 µM). In other words, there was a broad spectrum of activity. It is notable that [C_4_(MIM)_2_][Glu] had an MIC of 78 µM against *K. pneumonae*. Considering the lack of new antibiotics coming onto the market, and the threat caused, principally, by Gram-negative bacteria such as *K pneumonae* and *E. coli*, substances that have a broad antibacterial spectrum—e.g., [C_4_(MIM)_2_][Adi] and [C_10_(MIM)_2_][Pim]—and, in particular, some level of activity against *K. pneumonae*—e.g., [C_4_(MIM)_2_][Glu]—are always worthy of further study.

The results were in accordance with previous results reported for monocationic imidazolium-based ILs [33], which shows that the increase in the cationic alkyl chain for C_n_(MIM)Br (*n* = 8, 10, 12, and 14) leads to lower MICs. The strains evaluated (S*. aureus, E. coli, K. pneumoniae, and P. aeruginosa*) were resistant to C_8_MIM; however, all had their growth inhibited by C_14_MIM. Some DILs with spacers that have four and six carbons were active against *S. aureus* (ATCC 33591), *E. coli* (ATCC 25922), and *E. faecalis* (ATCC 29212) at lower concentrations (<1 mM), when possessing anions with long spacers (e.g., glutarate, adipate, and pimelate). Most DILs were active against *S. aureus* (ATCC 33591) at lower concentrations (<1 mM). These results follow the same trend as those previously reported for DILs.

Gindri et al. [10] evaluated the antibacterial activity that dicationic imidazolium-based ILs ((C_n_(MIM)_2_)^2+^) with amino acid-based anions had against *E. faecalis* (20–79 mM) and *P. aeruginosa* (5–156 mM) strains [10]. The authors observed the best activity against these strains to be hydrophobic ILs with the dicationic portion possessing ten methylene spacers.

Most of the DILs tested herein had some level of activity (MICs < 1.25 mM) against *S. aureus*. These results can be explained by the insertion of the hydrophobic chains of these ILs into the porous cell wall of the bacteria [30]. Results have shown that the insertion of functional groups in the cation (e.g., hydroxyl, esters, and amides) enhance the antimicrobial activity of imidazolium-based ILs [6], which is supported by the hypothesis that stronger electrostatic interaction results in higher antibacterial activity [34]. Considering this hypothesis and the results observed, two orthogonal effects (electrostatic interaction and hydrophobicity) might be competing against each other, given that DILs are double-charged ILs. Furthermore, cationic and anionic moieties have alkyl spacers, which makes these ILs hydrophobic.

Finally, there was no clear correlation between the length of the alkyl chain spacer of the cation and/or anion and its efficiency in inhibiting the bacteria growth, except for the results against *S. aureus*, and there was no systematic correlation between the alkyl chain spacer of the cations or dicarboxylate anions, and antibacterial activity against Gram-negative and Gram-positive cell membranes.

### 3.3. Toxicity Tests

*Caenorhabditis elegans* (*C. elegans*) has been used as a multicellular animal in toxicity tests, due to its small size, short life cycle, and ease of cultivation [18]. In contrast to in vitro tests, trials with *C. elegans* provide data from an animal with digestive, endocrine, sensory, neuromuscular, and reproductive systems. Additionally, about 40% of the genes of this nematode have apparent counterparts in humans [35]. For toxicological effects, imidazolium-based DILs were evaluated regarding the survival of *C. elegans*. The nematodes in their larval phase (more sensitive) were exposed to four concentrations (0.01, 0.1, 1.0, and 10 mM) of DILs for 1 h (Table 6).

Toxicity tests using *C. elegans* showed that in all the DILs, the nematode survival rate was above 75% at 0.01 and 0.1 mM. The ILs with cations [C_4_(MIM)_2_]^2+^ exhibited toxicity at 10 mM, regardless of the size of the anion spacer chain, except for [C_4_(MIM)_2_[Mal], which had no toxicity at any concentration. For [C_4_(MIM)_2_[Oxa], there was an increase in toxicity as the chain length and concentration increased. The [C_6_(MIM)_2_]^2+^ series had low toxicity even at 10 mM. Meanwhile, there was a decrease in nematode survival rate at higher concentrations of ILs with cations [C_8_(MIM)_2_]^2+^, except for [C_8_(MIM)_2_][Glu], which, even at 10 mM, had a survival rate of 90%. The ILs with cations [C_10_(MIM)_2_]^2+^ had the highest toxicity among the DILs tested, even at 0.01 mM. The nematode survival rate was less than 90% for [C_10_(MIM)_2_[Suc] and [C_10_(MIM)_2_[Glu]. The toxicity increased abruptly at the 10 mM concentration, with survival less than 70% for all DILs (Figure 4 and Figure 5).

These results for the cationic effect on *C. elegans* survival follow a trend that has already been reported by others studying monocationic ILs, in which, for the ILs [C_4_(MIM)][Cl], [C_8_(MIM)][Cl], and [C_14_(MIM)][Cl], the alkyl chain elongation leads to lethality against *C. elegans* at all concentrations (1.0–5.0 mg·mL^−1^). [C_4_(MIM)][Cl] did not affect the survival at any concentration, while [C_14_(MIM)][Cl] was lethal against *C. elegans*. This fact may be associated with the more lipophilic nature of ILs with longer chains, or because smaller chains can be eliminated more easily by the excretory system [36]. The toxic effects of the [C_8_(MIM)_2_][2X] DILs (in which X = Br, BF_4_, NO_3_, SCN, and Cl) on the survival of *C. elegans* was previously reported, with the worms’ survival rate above 90%, thus indicating no significant toxicity for these ILs at 0.01–10 mg·L^−1^ [37]. Therefore, considering these results and the ones presented herein, it can be seen that the toxicity of DILs against *C. elegans* was lower than that of monocationic IL, which may be associated with the additional cationic moiety. Thus, it is possible to use higher concentrations of ILs as antimicrobials without triggering toxic effects against healthy cells. It seems that the alkyl chain is “trapped” between the two cationic imidazolium heads and loses the ability to interact with the *C. elegans* membranes, thus reducing its toxicity.

However, the anion effect also appears to influence toxicity against *C. elegans*. Previous work showed that at low concentrations (0.01 mg·L^−1^), the ILs [C_8_C_2_(MIM)][Br], [C_10_C_2_(MIM)][Br], and [C_12_C_2_(MIM)][Br] had a hormetic effect on *C. elegans* [38]. The LC_50_ values against *C. elegans* were evaluated for 30 imidazolium-based monocationic ILs possessing several anions, side chains, and substitutions at the C2 of the imidazole [38]. The LC_50_ values for ILs with bromide and chloride decreased when lengthening the side chain, which indicates that longer chains enhance toxicity. Additionally, the ILs with bromide anions had greater acute toxicity than the ILs with chloride anions [38]. Furthermore, Wu et al. [27] showed that imidazolium-based monocationic ILs with bromide anions have dose-dependent toxic effects on *C. elegans*, with longer alkyl chains resulting in higher toxicity. Peng et al. [39] found no adverse effects on the growth, development, and locomotion of *C. elegans* after 72 h in the presence of [C_14_(MIM)][Br], at concentrations of either 5 or 10 mg·L^−1^. Considering these results for monocationic ILs and the ones presented herein for DILs, the influence of the anion decreases as the alkyl chain lengthens, which suggests that the cationic moiety has a dominant effect on the toxicity of monocationic ILs. It is worth noting that the toxic effect that the DILs presented herein showed against *C. elegans* was lower than that for most of the mono- and dicationic IL analogs with other anions.

## 4. Conclusions

In summary, our results showed that DILs with ten-carbon spacers were active against the *Candida* bacterial strains tested. By contrast, DILs with spacers shorter than ten carbons were active against some specific *Candidas* and bacteria (mainly *S. aureus*). The best activity against Gram-positive bacteria was observed for [C_4_(MIM)_2_][Pim], which had an MIC of 78 µM toward MRSA and also had a broad spectrum of activity. Most of the DILs are not cytotoxic at the MIC but can be cytotoxic at 1 mM. The range of antimicrobial activity and the cytotoxic/toxic concentration of the DILs described here was lower than for most of the DIL analogs with other anions described in the literature.

Additionally, the toxic effects that the DILs described herein showed against *C. elegans* were lower than for most of the mono- and dicationic ILs analogs with other anions. The influence of the anion decreases as the alkyl chain lengthens, which suggests that the cationic moiety has a dominant effect on the toxicity of monocationic ILs.

The results described herein showed that the presence of carboxylate anions reduces the toxicity of ILs. Additionally, the introduction of an organic natural anion leads to a decrease in toxic effects of imidazolium-based ILs compared with those containing halide anions, which is particularly important in the approach for designing novel, safer forms of biologically active compounds. Thus, we suggest that, if the right choice of concentration is made, some of the DILs described herein can be used in applications with live tissue; whereas others are more suitable as herbicidal, fungicidal, or even feeding deterrents toward stored product pests.

## Figures and Tables

**Figure 1 pharmaceutics-13-00639-f001:**
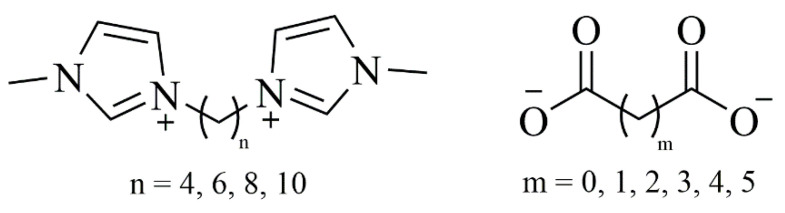
Molecular structure of dicationic ionic liquids.

**Figure 2 pharmaceutics-13-00639-f002:**
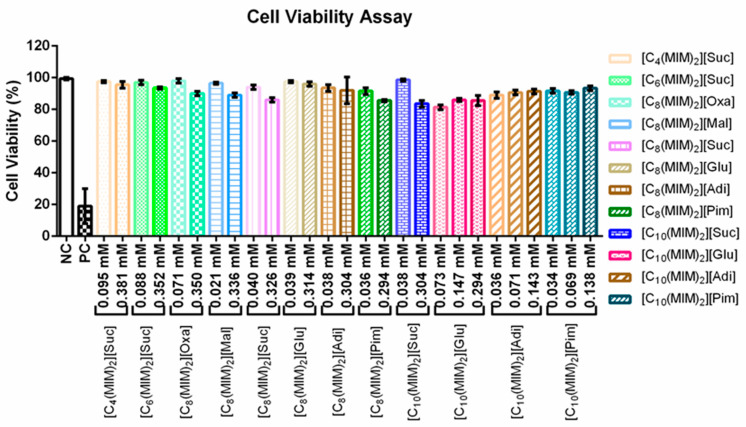
Cell viability evaluated by trypan blue assay after treatment with different concentrations (mM) of the ILs. Values are mean ± SD and indicate statistical differences compared by one-way ANOVA followed by the Tukey post hoc test. The lowercase letters indicate that *p* < 0.05 when compared to the corresponding control group. The mean values labeled with the same letter are not statistically different from each other.

**Figure 3 pharmaceutics-13-00639-f003:**
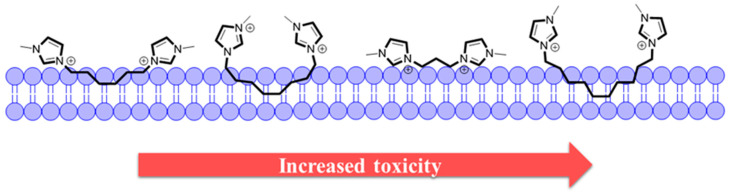
Interaction of DILs with the lipid bilayer by anchoring ([C_n_(MIM)_2_]^2+^, *n* = 4, 6, 8, 10. Anions have been omitted for better representation).

**Figure 4 pharmaceutics-13-00639-f004:**
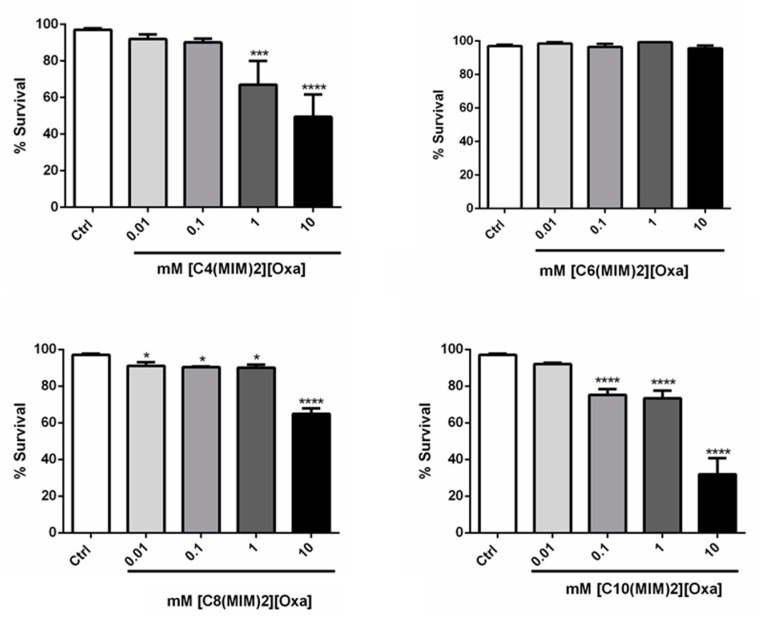
Survival assay for wild-type (N2) worms in different concentrations (in mM) of DILs with oxalate anion. * *p* < 0.05, *** *p* < 0.001 and **** *p* < 0.0001 compared to Ctrl (One-Way ANOVA fallowed by post-hoc Sidak).

**Figure 5 pharmaceutics-13-00639-f005:**
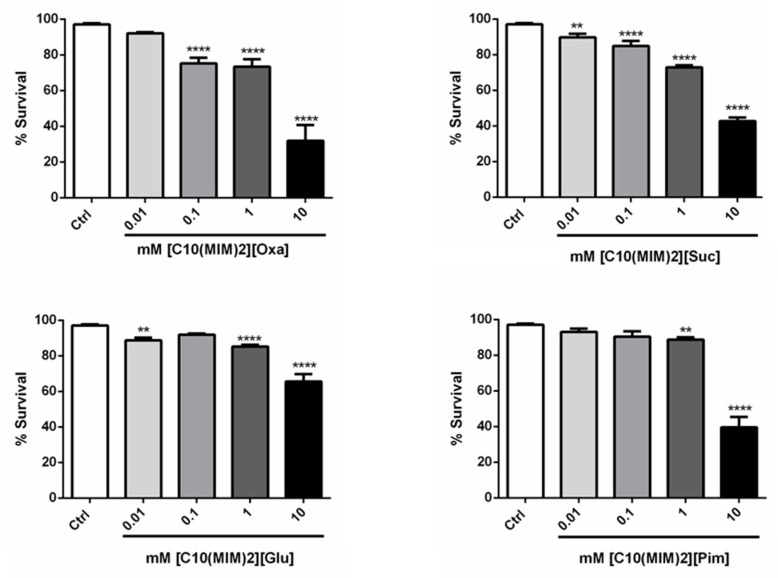
Survival assay for wild-type (N2) worms in C_10_(MIM)_2_[Oxa], C_10_(MIM)_2_[Suc], and C_10_(MIM)_2_[Oxa] C_10_(MIM)_2_[Pim]. ** *p* < 0.01 and **** *p* < 0.0001 compared to Ctrl (One-Way ANOVA fallowed by post-hoc Sidak).

**Table 1 pharmaceutics-13-00639-t001:** MICs (in mM) of DILs and Fluconazole against *Candidas*, after 24 h.

IL	*C. albicans* CA02	*C. krusei* CK03	*C. parapsilosis* CP RL38	*C. tropicalis* CT 08
[C_8_(MIM)_2_][Oxa]	>0.312	0.039	-	>0.312
[C_8_(MIM)_2_][Mal]	>0.312	>0.312	>0.312	0.039
[C_8_(MIM)_2_][[Suc]	>0.312	0.039	>0.312	0.078
[C_8_(MIM)_2_][Glu]	0.156	0.039	0.156	0.078
[C_8_(MIM)_2_][Adi]	>0.312	0.078	>0.312	0.039
[C_8_(MIM)_2_][[Pim]	>0.312	0.078	>0.312	0.039
Fluconazole	0.013	0.013	0.026	0.013

(-): not show growth.

**Table 2 pharmaceutics-13-00639-t002:** MICs (in mM) of DILs against *Candidas*, after 24 and 48 h.

IL	*C. albicans* CA02		*C. krusei* CK03		*C. parapsilosis* CP RL38		*C. tropicalis* CT 08	
	24 h	48 h	24 h	48 h	24 h	48 h	24 h	48 h
[C_4_(MIM)_2_][Suc]	0.156	>0.312	0.078	>0.312	0.156	>0.312	>0.312	>0.312
[C_6_(MIM)_2_][Suc]	0.039	0.078	0.078	0.156	0.078	>0.312	0.039	0.078
[C_8_(MIM)_2_][Suc]	>0.312	>0.312	0.039	0.078	>0.312	>0.312	0.078	>0.312
[C_10_(MIM)_2_][Suc]	0.078	0.078	0.039	0.039	0.039	>0.312	0.039	0.156

**Table 3 pharmaceutics-13-00639-t003:** log P values calculated by different platforms.

IL	log P		
	ChemDraw ^a^	Molinspiration (miLogP) ^b^	
[C_4_(MIM)_2_][Oxa]	dianion: −0.85	dication: −5.32	dianion: −3.76
[C_4_(MIM)_2_][Mal]	dianion: −0.94	dication: −5.32	dianion: −3.49
[C_4_(MIM)_2_][Suc]	dianion: −0.48	dication: −5.32	dianion: −3.22
[C_4_(MIM)_2_][Glu]	dianion: −0.03	dication: −5.32	dianion: −2.71
[C_4_(MIM)_2_][Adi]	dianion: 0.43	dication: −5.32	dianion: −2.21
[C_4_(MIM)_2_][Pim]	dianion: 0.89	dication: −5.32	dianion: −1.70
[C_6_(MIM)_2_][Oxa]	dianion: −0.85	dication: −5.00	dianion: −3.76
[C_6_(MIM)_2_][Mal]	dianion: −0.94	dication: −5.00	dianion: −3.49
[C_6_(MIM)_2_][Suc]	dianion: −0.48	dication: −5.00	dianion: −3.22
[C_6_(MIM)_2_][Glu]	dianion: −0.03	dication: −5.00	dianion: −2.71
[C_6_(MIM)_2_][Adi]	dianion: 0.43	dication: −5.00	dianion: −2.21
[C_6_(MIM)_2_][Pim]	dianion: 0.89	dication: −5.00	dianion: −1.70
[C_8_(MIM)_2_][Oxa]	dianion: −0.85	dication: −4.54	dianion: −3.76
[C_8_(MIM)_2_][Mal]	dianion: −0.94	dication: −4.54	dianion: −3.49
[C_8_(MIM)_2_][Suc]	dianion: −0.48	dication: −4.54	dianion: −3.22
[C_8_(MIM)_2_][Glu]	dianion: −0.03	dication: −4.54	dianion: −2.71
[C_8_(MIM)_2_][Adi]	dianion: 0.43	dication: −4.54	dianion: −2.21
[C_8_(MIM)_2_][Pim]	dianion: 0.89	dication: −4.54	dianion: −1.70
[C_10_(MIM)_2_][Oxa]	dianion: −0.85	dication: −3.70	dianion: −3.76
[C_10_(MIM)_2_][Mal]	dianion: −0.94	dication: −3.70	dianion: −3.49
[C_10_(MIM)_2_][Suc]	dianion: −0.48	dication: −3.70	dianion: −3.22
[C_10_(MIM)_2_][Glu]	dianion: −0.03	dication: −3.70	dianion: −2.71
[C_10_(MIM)_2_][Adi]	dianion: 0.43	dication: −3.70	dianion: −2.21
[C_10_(MIM)_2_][Pim]	dianion: 0.89	dication: −3.70	dianion: −1.70

^a^ ChemDraw12 was unable to calculate the log P value for DILs or dications separately. ^b^ The Molinspiration tool does not allow the calculation of the DIL, only the separate dication and dianion.

**Table 4 pharmaceutics-13-00639-t004:** MICs (in mM) of DILs against Gram-positive bacteria.

IL	*S. aureus* ^a^	*S. aureus* ^b^	*E. faecalis* ^c^	*E. faecalis* ^d^
[C_4_(MIM)_2_][[Oxa]	>2.5	>2.5	>2.5	>2.5
[C_4_(MIM)_2_][Mal]	>2.5	>2.5	>2.5	>2.5
[C_4_(MIM)_2_][Suc]	1.25	2.5	>2.5	>2.5
[C_4_(MIM)_2_][Glu]	2.5	1.25	>2.5	>2.5
[C_4_(MIM)_2_][Adi]	1.25	1.25	0.625	0.625
[C_4_(MIM)_2_][Pim]	0.625	0.078	>2.5	>2.5
[C_6_(MIM)_2_][Oxa]	>2.5	2.5	>2.5	>2.5
[C_6_(MIM)_2_][Mal]	0.312	1.25	>2.5	>2.5
[C_6_(MIM)_2_][Suc]	2.5	1.25	>2.5	>2.5
[C_6_(MIM)_2_][Glu]	2.5	>2.5	>2.5	>2.5
[C_6_(MIM)_2_][Adi]	0.132	1.25	>2.5	>2.5
[C_6_(MIM)_2_][Pim]	>2.5	>2.5	>2.5	>2.5
[C_8_(MIM)_2_][Oxa]	1.25	0.312	1.25	>2.5
[C_8_(MIM)_2_][Suc]	0.156	1.25	>2.5	>2.5
[C_8_(MIM)_2_][Glu]	0.312	2.5	1.25	>2.5
[C_8_(MIM)_2_][Pim]	2.5	1.25	>2.5	>2.5
[C_10_(MIM)_2_][Glu]	2.5	>2.5	>2.5	>2.5
[C_10_(MIM)_2_][Pim]	0.625	0.625	1.25	1.25
Control (µg/mL)	2	2	2	2
Control (µM)	5.72	5.72	5.72	5.72

^a^ ATCC 25923, ^b^ ATCC 33591, ^c^ ATCC 29212, ^d^ ATCC 51299. Control: AMPICILIN (CLSI 2016) for *S. aureus* and *E. faecalis*.

**Table 5 pharmaceutics-13-00639-t005:** MICs (in mM) of DILs against Gram-negative bacteria.

IL	*E.* *Coli* ^a^	*K.* *pneumoniae* ^b^	*A.* *baumannii* ^c^	*A.* *baumannii* ^d^	*P.* *Aeruginosa* ^e^	*E.* *faecium* ^e^	*E.* *aerogenes* ^f^
[C_4_(MIM)_2_][[Oxa]	>2.5	>2.5	>2.5	>2.5	>2.5	>2.5	>2.5
[C_4_(MIM)_2_][Mal]	>2.5	>2.5	>2.5	>2.5	>2.5	>2.5	>2.5
[C_4_(MIM)_2_][Suc]	2.5	>2.5	>2.5	>2.5	>2.5	>2.5	>2.5
[C_4_(MIM)_2_][Glu]	>2.5	0.078	>2.5	>2.5	>2.5	>2.5	>2.5
[C_4_(MIM)_2_][Adi]	0.312	>2.5	>2.5	>2.5	2.5	0.625	>2.5
[C_4_(MIM)_2_][Pim]	>2.5	>2.5	>2.5	>2.5	>2.5	>2.5	>2.5
[C_6_(MIM)_2_][Oxa]	>2.5	>2.5	>2.5	>2.5	>2.5	>2.5	>2.5
[C_6_(MIM)_2_][Mal]	1.25	>2.5	>2.5	>2.5	2.5	>2.5	>2.5
[C_6_(MIM)_2_][Suc]	>2.5	>2.5	>2.5	>2.5	>2.5	>2.5	>2.5
[C_6_(MIM)_2_][Glu]	>2.5	>2.5	>2.5	>2.5	>2.5	>2.5	>2.5
[C_6_(MIM)_2_][Adi]	>2.5	>2.5	>2.5	>2.5	>2.5	>2.5	>2.5
[C_6_(MIM)_2_][Pim]	1.25	>2.5	>2.5	>2.5	>2.5	>2.5	>2.5
[C_8_(MIM)_2_][Oxa]	2.5	>2.5	>2.5	1.25	>2.5	>2.5	>2.5
[C_8_(MIM)_2_][Suc]	1.25	>2.5	>2.5	>2.5	>2.5	>2.5	>2.5
[C_8_(MIM)_2_][Glu]	0.625	0.312	>2.5	0.156	>2.5	>2.5	0.156
[C_8_(MIM)_2_][Pim]	2.5	>2.5	>2.5	>2.5	>2.5	>2.5	>2.5
[C_10_(MIM)_2_][Glu]	>2.5	>2.5	>2.5	>2.5	>2.5	>2.5	>2.5
[C_10_(MIM)_2_][Pim]	0.625	0.625	>2.5	>2.5	0.312	0.156	0.156
Control µg/ml	4	2	2	2	1	2	1
Control (µM)	11.44	5.72	6.30	3.15	3.15	5.72	3.15

^a^ ATCC 25923, ^b^ ATCC 700603, ^c^ ATCC 19606, ^d^ IOC 3174, ^e^ ATCC 27853, ^f^ ATCC 13048. Controls: AMPICILIN (CLSI 2016) for *E. coli*, *K.*
*pneumoniae,* and *E. faecium*; and *IMIPENEN* for *A. baumannii* and *P. aeruginosa.*

**Table 6 pharmaceutics-13-00639-t006:** DIL survival assay for wild-type (N2) worms (survival ± SD) ^a^.

IL	Concentration (mM)			
	0.01	0.1	1	10
[C_4_(MIM)_2_][Oxa]	92 ± 6	90± 4	67± 26 ***	50 ± 27 ****
[C_4_(MIM)_2_][Mal]	97 ± 2	98 ± 3	98 ± 2	94 ± 4
[C_4_(MIM)_2_][Suc]	92 ± 6	92 ± 7	92 ± 8	76 ± 10 ****
[C_4_(MIM)_2_][Glu]	96 ± 5	92 ± 4	89 ± 8	82 ± 11 ****
[C_4_(MIM)_2_][Adi]	91 ± 8	92 ± 9	90 ± 8	86 ± 4 **
[C_4_(MIM)_2_][Pim]	95 ± 5	89 ± 6 *	84 ± 8 ***	89 ± 8 *
[C_6_(MIM)_2_][Oxa]	99 ± 2	97 ± 4	99 ± 2	96 ± 5
[C_6_(MIM)_2_][Mal]	99 ± 2	95 ± 4	96 ± 4	96 ± 4
[C_6_(MIM)_2_][Suc]	97 ± 4	98 ± 2	93± 5	95 ± 3
[C_6_(MIM)_2_][Glu]	100 ± 0	99 ± 2	97± 3	98± 2
[C_6_(MIM)_2_][Adi]	95 ± 2	100 ± 0	99 ± 3	96 ± 3
[C_6_(MIM)_2_][Pim]	98 ± 2	96 ± 4	98 ± 1.72	97 ± 3
[C_8_(MIM)_2_][Oxa]	91 ± 4 *	90 ± 1 *	90± 3 *	65 ± 5 ***
[C_8_(MIM)_2_][Mal]	92 ± 4	78 ± 8 ****	64 ± 12 ****	42 ± 11 ****
[C_8_(MIM)_2_][Suc]	93 ± 2	92 ± 3	85 ± 2 *	19 ± 15 ****
[C_8_(MIM)_2_][Glu]	93 ± 1	92 ± 3	95 ± 2	90 ± 4
[C_8_(MIM)_2_][Adi]	92 ± 3	79 ± 7 ****	74 ± 8 ****	30 ± 9 ****
[C_8_(MIM)_2_][Pim]	92 ± 4	91 ± 6	75 ± 5 ****	18 ± 17 ****
[C_10_(MIM)_2_][Oxa]	92 ± 1	75 ± 6 ****	74 ±7 ****	32 ± 15 ****
[C_10_(MIM)_2_][Suc]	90 ± 4 **	85 ± 5 ****	73 ± 2 ****	43 ± 4 ****
[C_10_(MIM)_2_][Glu]	89 ± 3 **	92 ± 1	85 ± 2****	66 ± 7 ****
[C_10_(MIM)_2_][Pim]	93 ± 3	90 ± 5	89 ± 2 **	40 ± 10 ****

^a^ Control (0 mM): Survival ± SD = 97.01 ± 3.32. Data are expressed as percentage of living worms from four independent assays of approximately 100 worms per group in each experiment (*n* ± 3, with six experiments per group). Error bars represent the means ± S.E.M. * *p* < 0.05, ** *p* < 0.01, *** *p* < 0.001, and **** *p* < 0.0001 compared to the control (one-way ANOVA followed by post-hoc Sidak).

## Data Availability

Not applicable.

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
