# Peer review of "Antimicrobial and Toxicity Evaluation of Imidazolium-Based Dicationic Ionic Liquids with Dicarboxylate Anions"

_pharmaceutics, 2021, doi:10.3390/pharmaceutics13050639_

Round 1

Reviewer 1 Report

The authors made this study very carefully and conducted toxicity experiments with the new class of ionic liquids. The synthesis and characterization of DILs were given in their previous work, however, a brief description and chemical reaction of the synthesis should be given in this paper. It would be much more convenient for the reader to follow the text. Also, the quality of the figures should be improved and in accordance with the journal requirements. 

It is well known that oxygen atoms in ILs reduce the toxicity of the ionic liquids, thus it was expected to obtain lower toxicity by the introduction of dicarboxylic anions. But what is the role of the dications in the overall toxicity of DILs? Can the authors evaluate it and discuss it in the manuscript?

Author Response

Dear Reviewer, 

We appreciate your comments  and are thankful for the analysis of our manuscript and for the insightful comments. We tried to address each of the questions and suggestions. We hope that the revised version of the manuscript and clarifications provided in this letter will address the concerns raised by you.

 Sincerely,

Clarissa P. Frizzo

Reviewer 1: The authors made this study very carefully and conducted toxicity experiments with the new class of ionic liquids.

Response: We appreciate the reviewer’s comments.

Comment 1. The synthesis and characterization of DILs were given in their previous work, however, a brief description and chemical reaction of the synthesis should be given in this paper. It would be much more convenient for the reader to follow the text.

Response: The experimental procedure of the synthesis od DILs were added in the manuscript. (lines 108-118)

Comment 2. Also, the quality of the figures should be improved and in accordance with the journal requirements.

Response: The quality of the figures was improved in accordance with the journal requirements.

Comment 3. It is well known that oxygen atoms in ILs reduce the toxicity of the ionic liquids, thus it was expected to obtain lower toxicity by the introduction of dicarboxylic anions. But what is the role of the dications in the overall toxicity of DILs? Can the authors evaluate it and discuss it in the manuscript?

Response: The role of the dications on the toxicity of DILs was evaluated in the manuscript. The evidence for that can be found in some parts of manuscript: (i) Introduction (lines 48-55), where we highlight that dicationic ILs (DILs) are less toxic than their monocationic analogues; (ii) Results (lines 234-257) where we introduce a  discussion about cytotoxicity of DILs compared with monocationic ILs and a discussion about the effect of length of alkyl chain of DILs in citotoxicity (See Figure 3). (iii) Results (lines 389-422), where we discuss the effects of the cation nature on the toxicity against C. elegans. Similar to citotocicity tests, the comparison between mono- and dicationic ILs and effect of lengh of alkyl chain on toxicity was introduced.

Particularly in lines (240-243), we cited that the insertion of an additional imidazole head make the DILs more polar than monocationic ILs, which leads to a reduction in toxicity.

Reviewer 2 Report

In this work, the authors studied imidazolium (DIL) for use in living tissues to prevent bacterial or fungal infections. To carry out this study, the authors have determined the antibacterial activity, antifungal activity, and cell viability of different imidazolium-based DIL species.

The work seems to me of good quality, the introduction is well described and helps the reader to understand the problem and the approach of the authors on the subject in question. the methodology is well described. The data provided by the authors is clear and well described. the conclusions are clear and well argued.

clarifications

The authors should review the way of putting the citations in the text since the authors use two different ways of citing, for example line 74 and line 91.

table 6.

The authors should review the values expressed in this table, the values are well expressed but the errors (SD) cannot be stated with so many figures. the normal thing is to express errors with a significant figure only when this is a one, they can be expressed with two figures.

Author Response

Dear Reviewer,

We appreciate your comments  and are thankful for the analysis of our manuscript and for the insightful comments. We tried to address each of the questions and suggestions. We hope that the revised version of the manuscript and clarifications provided in this letter will address the concerns raised by you.

Sincerely,

Clarissa P. Frizzo

Reviewer 2: In this work, the authors studied imidazolium (DIL) for use in living tissues to prevent bacterial or fungal infections. To carry out this study, the authors have determined the antibacterial activity, antifungal activity, and cell viability of different imidazolium-based DIL species. The work seems to me of good quality, the introduction is well described and helps the reader to understand the problem and the approach of the authors on the subject in question. the methodology is well described. The data provided by the authors is clear and well described. the conclusions are clear and well argued.

Response: We appreciate the reviewer’s comments.

Clarifications

Comment 1. The authors should review the way of putting the citations in the text since the authors use two different ways of citing, for example line 74 and line 91.

Response: We have revised and corrected the citations.

Comment 2. Table 6. The authors should review the values expressed in this table, the values are well expressed but the errors (SD) cannot be stated with so many figures. the normal thing is to express errors with a significant figure only when this is a one, they can be expressed with two figures.

Response: the SD was corrected accordingly.